# Sensor-Aided EMF Exposure Assessments in an Urban Environment Using Artificial Neural Networks

**DOI:** 10.3390/ijerph17093052

**Published:** 2020-04-28

**Authors:** Shanshan Wang, Joe Wiart

**Affiliations:** Chaire C2M, LTCI, Télécom Paris, Institut Polytechnique de Paris, 91120 Palaiseau, France

**Keywords:** EMF exposure, radiofrequency, artificial neural network, locally-connected layers, EMF sensor network

## Abstract

This paper studies the time and space mapping of the electromagnetic field (EMF) exposure induced by cellular base station antennas (BSA) using artificial neural networks (ANN). The reconstructed EMF exposure map (EEM) in urban environment is obtained by using data from EMF sensor networks, drive testing and information accessible in a public database, e.g., locations and orientations of BSA. The performance of EEM is compared with Exposure Reference Map (ERM) based on simulations, in which parametric path loss models are used to reflect the complexity of urban cities. Then, a new hybrid ANN, which has the advantage of sorting and utilizing inputs from simulations efficiently, is proposed. Using both hybrid ANN and conventional regression ANN, the EEM is reconstructed and compared to the ERM first by the reconstruction approach considering only EMF exposure assessed from sensor networks, where the required number of sensors towards good reconstruction is explored; then, a new reconstruction approach using the sensors information combined with EMF along few streets from drive testing. Both reconstruction approaches use simulations to mimic measurements. The influence of city architecture on EMF exposure reconstruction is analyzed and the addition of noise is considered to test the robustness of ANN as well.

## 1. Introduction

The risk perception of electronic-magnetic field (EMF) exposure [1] is nowadays a hot issue with the fast development of wireless communications. The actual deployment of 5G infrastructures and base station antennas (BSA) strengthen such concern and reinforce the needs of EMF exposure assessment [2] and monitoring. In Europe, the EMF monitoring is often carried out using “one-time” measurement campaigns as described in [3,4]. Projects, like LEXNET [5], also studied population radiofrequency (RF) exposure. The monitoring can also be done using fixed sensors [6]. Recently, autonomic in energy and wide-band sensors have been proposed [7]. These sensors are installed and tested in cities such as Paris and Toulouse ( Observatoire des Ondes [8]). The sensor networks, as well as on-site measurements, are of great interest, but both methods are localized, which means EMF exposure is monitored at limited locations. Therefore, it is necessary to build an EMF Exposure Map (EEM) from these measurements.

In the present paper, the time and space mapping of outdoor EMF exposure induced by BSA in the 4G cellular networks is studied using artificial neural networks (ANN). The EEM in urban environment is reconstructed from sensor networks, drive testing, and information related to BSA that are accessible on the public database, including spatial locations and orientations of BSA. This information related to BSA is often used to design and optimize wireless networks but rarely to deal with exposure assessment and monitoring. Other EMF sources, e.g., high-voltage transmission lines, substation equipment, are not considered in the present paper.

However, there are several challenges, which prevent us from reconstructing an accurate spatial map of EMF exposure using artificial intelligence (AI). Due to the complexity of building structure and city architecture, the propagation encounters multiple reflection, refraction and diffraction, which induces highly varying EMF in space. The challenge is to convert and process information into readable input variables of ANN. It is also important to find the significant features in RF exposure instead of feeding redundant inputs into ANN.

The mentioned challenges bring significant uncertainty and hinder us from reconstructing the EEM. Conventional methods, e.g., ray-based simulators [9], network-based measurements [10], and recent approaches like Kriging [11,12], are used in assessing EMF exposure. However, these approaches are either too complicated in terms of costs or not able to take into account all the meaningful features. Considering the complexity, accuracy and the ability to deal with high-dimension data, the mentioned methods are not feasible to cover all the aspects.

Due to the limitation in the diversity of inputs and the flexibility of the methods, we are strongly motivated to use ANN, a promising and novel approach, to reconstruct the spatial map of EMF exposure. ANN is one of the trending approaches nowadays, because of its efficiency and universality. It has been widely applied in solving environmental and public health issues, e.g., air pollution monitoring [13] and water quality modeling [14]. Recently it was applied in wireless communications [15,16] to quantify interference or coverage performance and propagation path loss (PL) prediction. In [17,18,19], PL is assessed using ANN with consideration of distance, antenna and terrain information. However, unlike PL, EMF exposure involves much more BSA, and thus it is linked to the aggregation of all possible PL plus noise. If we adopt the approach used in PL assessment, which considers all active BSA as inputs of ANN, it would bring the high-dimension inputs problem.

Therefore, our team started to assess the EMF exposure using a simple ANN model [20]. However, the simple and general ANN model is not sufficient to handle the large variability of the exposure and complexity of the city.

In the present paper, we present the reconstruction of EEM using a proposed hybrid ANN model based on drive testing assisted by sensor networks (both are extracted from simulations) and information on BSA. To analyze the accuracy and the robustness of different ANN and reconstruction methods, various EMF Exposure Reference Maps (ERM) are calculated using simulations, where a parametric propagation PL, able to reflect the city complexity, is used. to be more specific, we have the following contributions:We propose a new hybrid ANN model, and compare its performance with conventional fully-connected ANN in reconstructing EMF exposure. In order to make better use of key inputs of ANN, including distance to the source, direction of the transmitting antennas, blockage in the surrounding environment, time variation and background noise, we propose a hybrid ANN. The idea is to process information from one BSA locally, then concentrate the data from different BSA together.We propose a new reconstruction method based on the one-time drive testing with the help of sensor networks. We compare its performance against the reconstruction method based on sensor networks only. The new reconstruction method has the advantage of achieving good predictions with lower cost.

The paper is organized as follows: Section 2 introduces the system model and settings, Section 3 introduces the general set up of the neural network and how it can be applied to solve regression problems. Section 4 proposes a new type of hybrid neural network. Section 5 describes two main approaches used to reconstruct RF exposure. Section 6 validates the propositions in the previous sections and compares the performance between different ANN models with and without consideration of noise. Section 8 concludes the paper.

## 2. System Model

### 2.1. Exposure Reference Map Construction and Setting

In this paper, a 4G cellular network with BSA operating at 2600 MHz is considered. The real geographical locations of BSA used in the simulations are available on ANFR’s databases [21]. The outdoor EMF exposure in the 14th district of Paris is of interest in the present paper, which can be interpreted as the EMF exposure received at the soil level of the open streets. The map of 14th district is shown in Figure 1, with real spatial locations of BSA and street lamps displayed in black and red dots, respectively. Sensor networks are deployed on selected street lamps and actively monitoring the real-time EMF exposure. In total, there are 3516 street lamps in the 14th district. The specifications of the sensor networks can be found in [7].

Considering that each BSA is equipped with one-directional antenna operating at 2600 MHz, it is assumed that:-the direction of each antenna is uniformly distributed;-the transmit power for each BSA is equal.

In order to capture the diverse and varying blockages, the following PL model is proposed.

### 2.2. Parametric Exposure Reference Map

The EMF emitted from a source can be approximated as a spherical wave when the observer is in the far field region and free space. Therefore the received signal strength [22] is inversely proportional to the distance between the source and the receiver. The simple propagation PL model is widely-used [23]:(1)$$E=B+10\alpha \log_{10}\left(d\right)+s;$$
where *E* is the electromagnetic field, *B* is a constant, *d* is the distance to the source and *s* is the shadow fading variation. α is the Path Loss Exponent (PLE).

PLE is used to take into account the environment in the propagation. EMF are reflected, refracted and diffracted by obstacles on the propagation path. Each link between the receiver and the BSA experiences different attenuation caused by obstacles in the propagation, which can be represented by PLE. However, a simple PL model with a fixed value of PLE has limitations in representing complicated urban environment. In order to better mimic the varying propagation channel, we use the following block-based stochastic PL model [9]. In this case, PLE is not a deterministic value anymore, but a random variable governed by the statistical law.

#### Stochastic Block-Based Path Loss Model

Since the BSA are usually located at high positions, e.g., rooftops or tower tops, the receivers’ surroundings are more complicated and important. Therefore, we use a block-based PL model. In this approach PLE $\alpha_{x_{j}}$ is dependent on the location of receiver $x_{j}$, which is further determined by surrounding blockages around $x_{j}$ in the environment.

Here, block-based means different regions may have different surrounding environment, e.g., locations around open space, like a green land or a square, would have a higher Line-of-Sight (LoS) probability and therefore the value of PLE is close to 2. On the contrary, locations among buildings or skyscrapers would be more likely to have high PLE values. In Figure 2 (left), the 14th district is divided into four blocks, where each block represents a different averaged PLE denoted by a different color. In the same block, the receivers may experience slightly different structures, and simultaneous mobile objects. Therefore, Figure 2 (middle) shows all PLE $\alpha_{x}$ for locations of receivers *x*, which are generated from normal distribution $N\left(\mu ,\sigma^{2}\right)$, where μ is given by the mean PLE of the block. Figure 2 (right) shows PLE distribution for receivers located in α=3.5 block.

**Remark** **1.**
*This division of the region into different blocks is not obtained from empirical measurements. This assumption provides a solution towards reproducing diverse PLE level, which depends on the obstacles in the environment. Other irregular blocks or clusters can be used as well, based on the empirical environments. Figure 2 shows a non-empirically-based example of a parametric setup for PLE. None of the PLE values are obtained from empirical values nor real city structure. The goal is to demonstrate how the parametric path loss model works.*


**Remark** **2.**
*Here in the present work, we do not consider the shadowing effect in the Equation (Equation 1), however, the block-based PL model is able to take it into account. The variations of PLE for receivers located in the same block can be interpreted as shadowing. The level of ‘shadowing’ can be controlled by tuning $\sigma^{2}$ in normal distribution.*


The received EMF follows the PL model denoted in Equation (Equation 1). This block-based PLE model follows the rule: no matter where the BSA is located, if the receiver is located at the block with certain PLE, this PLE is adopted in the PL model.

### 2.3. Time Variation

Due to the time variation of the traffic load, the total power transmitted by all BSA varies as well, which leads to the varying exposure levels at different times of day, especially in urban environments. Usually, the peak value of exposure is experienced around noon while the bottom is reached around dawn every day. A typical wide-band time variation pattern can be found in Figure 3, from one sensor in an assay deployed by EXEM [7] in the 14th district of Paris. This periodical phenomenon shows a significant peak and trough around noon and dawn in a day, respectively.

In the present paper, we use a trigonometric function to model the time variation pattern in the selected frequency band, since it can capture the most important features in the periodic pattern. Equation (Equation 2) gives the formula of exposure in terms of EMF after considering time variation.
(2)$${\tilde{E}}_{rx}=E_{rx}*f_{t}\left(t\right),$$
where $f_{t}\left(t\right)=-0.4*sin\left(t\right)+1$.

**Remark** **3.**
*Here, we assume the same time variation pattern for all BSA. Then the total EMF exposure with time variation can be simplified as previous exposure multiplied by the time variation function as seen in Equation (Equation 2). The sensor networks can record exposure for a whole day while the drive testing is carried out during working hours.*


### 2.4. Adding Noise to Exposure

As a vital part of wireless communication, the existence of background noise cannot be neglected. It includes thermal noise from electronic devices and signals originated from unlicensed frequency bands or distanced channels [24].

The additive white Gaussian noise (AWGN) is taken into account when we simulate the outdoor exposure. In addition, the noise is added to test the robustness of the neural network. Results from noiseless and with noise cases are compared and evaluated in Section 6.

## 3. Regression Neural Network Set Up

An ANN aimed at solving regression problems can be constructed as shown in Figure 4. Inputs $x_{1},x_{2},\ldots ,x_{N}$ are selected from possible influential factor resulting in EMF exposure. Here, $x_{0}$ is added as bias. The biases are added as an additional “weight” in each hidden layer, so that to adjust the output along with the weighted sum of the inputs to the neuron. The concept is similar but not limited to the intercept added in a linear regression. It should be noticed that this term is different from the idea of a statistical bias, in which a statistical estimation of the algorithm’s expected estimate of a quantity is not equal to the true quantity.

On the ANN in this paper, many factors contribute to the received signal strength at a certain location, e.g., the distance between the emitting and receiving antennas, their directions as well as the city structure itself. After the inputs are fed to the ANN, it processes the information extracted from the inputs through hidden layers, where hidden layers are composed of neurons. Different hidden layers are connected by weights, e.g., neuron $a_{i}^{\left(j\right)}$ from the *j*-th layer can be obtained by the following formula:(3)$$a_{i}^{\left(j\right)}=g\left(\theta_{i0}^{\left(j-1\right)}a_{0}^{\left(j-1\right)}+\theta_{i1}^{\left(j-1\right)}a_{1}^{\left(j-1\right)}+\ldots +\theta_{in}^{\left(j-1\right)}a_{n}^{\left(j-1\right)}\right),$$
where $g\left(x\right)$ represents the activation function. In the present paper, the activation function for all hidden layers is refined linear unit (reLU), except for the output layer, where a pure linear function is used. *n* is the number of neurons in $\left(j-1\right)$-th hidden layer.

After the forward propagation is done, the cost function is constructed as:(4)$$J\left(\theta \right)=-\frac{1}{m}\left[\sum_{i=1}^{m}\sum_{k=1}^{K}y_{k}^{\left(i\right)}\log\left(h_{\theta }{\left(x^{\left(i\right)}\right)}_{k}\right)+\left(1-y_{k}^{\left(i\right)}\right)\log\left(1-h_{\theta }{\left(x^{\left(i\right)}\right)}_{k}\right)\right],$$
where K represents the total number of outputs that $y^{\left(i\right)}$ has, *m* is the total number of training examples.

Then, to minimize the cost function in Equation (Equation 4), back propagation is performed by applying gradient descend method. To better evaluate the performance of the ANN, two metrics are used in the present paper, mean square error (MSE) and R^2^ [25]. MSE is used to minimize the residual sum of squares (RSS) and R^2^ indicates how close two sets of data are in terms of distribution. To be more specific, R^2^ is defined as:(5)$$R^{2}=1-\frac{RSS}{TSS},$$
where $TSS=\sum {\left(y_{i}-\bar{y}\right)}^{2}$ is the total sum of squares, and RSS is residual sum of squares, defined as $RSS=\sum_{i=1}^{m}{\left(y_{i}-\hat{y_{i}}\right)}^{2}$. TSS measures the total variance in the response Y. In contrast, RSS measures the amount of variability that is left unexplained after performing the regression. Consequently, when R${}^{2}\to 1$, a large proportion of the variability in the response has been explained by the regression [26].

### Tuning Hyper-Parameters

While the ANN is set up, to make the most use of the ANN, tuning the hyper-parameters becomes a vital problem. Usually, this tuning process requires not only the experience and skills of the user, but also the mastery of features of data. The grid search method is used in the present paper to select the optimal combination of hyper-parameters with help from [27].

## 4. New Hybrid-Connected Neural Networks

In this section, we propose a new ANN structure with not-fully connected layers. After inspecting the physical nature of EMF exposure, we found that the exposure is aggregated from all effective BSA in the receiving range. Each BSA has its specifications, e.g., the azimuth, the location and the elevation. Therefore, to imitate and reproduce the real propagation, locally-connected layers are constructed in parallel to process the inputs from the same BSA. Then, fully-connected layers are constructed to process the concentrated outputs from locally-connected layers.

Figure 5 shows the structure of the hybrid ANN, inputs $\left\{x_{1},x_{2}\right\}$ and $\left\{x_{P-1},x_{P}\right\}$ are processed in independent locally-connected blocks and there are no single mutual inputs in different blocks. Then, $\left\{x_{P+1},\ldots ,x_{N}\right\}$ as added raw inputs to fully-connected layers, together with outputs from locally-connected layers. Here, the physical meaning of $\left\{x_{P+1},\ldots ,x_{N}\right\}$ can be interpreted as information, like time, location of the receiver, which remains same for different BSA in one training sample.

**Remark** **4.**
*This new structure of ANN is proposed to suit to the specific scenario, where inputs are clustered and organized by each BSA. The output is summed from all effective BSA. This structure may also be applied to cases, where inputs and outputs have similar relationships. In [28], a similar semi-parallel structure in used in convolutional neural networks. The advantage of hybrid connected ANN is that, it reduces unnecessary interactions between some neurons, which reduces the complexity of the network and improves the computation efficiency.*


The proposed locally-connected ANN is not only capable of reproducing the predictions of exposure just as the conventional ANN, it can also do the job more efficiently. Validations and comparisons can be found in Section 6.

## 5. Reconstruction Methods

In this section, we explore two different reconstruction methods, in order to find balance between the cost of deploying sensors and the accuracy of predicting outdoor RF exposure.

### 5.1. Case 1: Through Sensor Networks

Since the sensor networks can continuously record EMF exposure for long time, the time variation pattern is fully available in the data recorded by the sensors. Reconstructing the spatial map of EMF exposure by sensor networks is first considered. Here, inputs of ANN consist of distances to 10 nearest BSA, azimuth of antennas, locations of receivers and time of the measurement.

In Figure 6, an example with 200 sensors selected from 3516 street lamps is given. The sensor based reconstruction method may bring a high cost problem in deploying real sensors. Therefore, in Section 6, the trade-off between required number of sensors and performance of the reconstruction is explored.

### 5.2. Case 2: Through Combined Drive Testing and Sensor Networks

Bearing in mind the potential high-cost limitation in sensor deployments, we propose a new reconstruction approach, which requires a lower number of sensors while maintaining a good reconstruction performance. We combine the one time drive testing and sensor networks together. Since the drive testing is done at different time of the day or several days and the peak of the periodic time variation usually occurs during noon in a typical day. This means that the drive testing may not be able to provide enough insights of the full-time variation pattern of the EMF exposure.

To be more specific, we keep the same BSA information available publicly as used in case 1, then, additional inputs extracted from sensor networks are considered as inputs to assist the reconstruction: the distance to nearest sensor and EMF exposure record by that sensor. The novelty of this reconstruction approach is the combination of all available information together, including sensors, drive testing and BSA information while less cost is required. The results comparing both methods can be found in Section 6.

## 6. Results

In this section, sustainable results obtained from ANN are presented to illustrate that the ANN is able to capture the main features in EMF exposure emitted from BSA. EMF exposure is reconstructed under the following two scenarios: (1) using sensor networks only, (2) combining drive testing and sensor networks.

The system model described in Section 2 is used in the simulation settings. Two scenarios with and without noise are compared in the results. Other system set up can be found in Table 1.

The construction and training of ANN is carried out using Python version 3.7.3 on a PC under Tensorflow environment (GPU implemented on Nvidia Quadro RTX 8000 seed-Architecture Turin [29] will be used in the future work). Main packages used in the present paper are Keras and Sklearn. The hyper-parameters used in the present paper are shown in Table 2. The initial weight is determined from Glorot uniform initializer [30], which draws samples from a uniform distribution within [-limit, limit]. Here, limit $=\sqrt{6/\left({\mathrm{fan}}_{in}+{\mathrm{fan}}_{out}\right)}$, where ${\mathrm{fan}}_{in}$ and ${\mathrm{fan}}_{out}$ represent the number of input units and output units in the weight tensor respectively. Biases are initialized to be zeros. Standardization is used to pre-process inputs. LC and FC represent locally-connected and fully-connected (layers).

In case 1, different percentages of training and testing are explored. We have inputs of the ANN as: (1) distance to 10 nearest BSA, (2) location of the receiver, (3) angle between receiver and direction of maximum gain of antenna, (4) time of the sensor measurements. In Figure 7, R${}^{2}$ and MSE performance are compared under conventional and hybrid ANN, validated by two different sets of PLE. The hybrid ANN always outperforms the conventional one with different percentages of training and testing. With different percentage of training and testing, we are able to achieve at most $R^{2}=0.80$ with 50% sensors (1758) required. When the number of training data decreases, the performance of the prediction also decreases as expected. In the worst case, where 50 sensors (1.4%) are used as training, the performance of R${}^{2}$ drops to less than 0.5 with high variation. Figure 8 and Figure 9 illustrate reconstructed EEM, ERM and absolute error maps under $N_{training}=\left\{50,1758\right\}$ cases, it is clear that decreasing training examples would degrade the prediction of ANN significantly. However, a deployment of a larger number of sensors would imply a higher cost for installation and maintenance, thus leading to an impractical situation.

In case 2, sensor information is used to assist prediction done by drive testing. In addition to the inputs we considered in case 1, we add two more information feeds as inputs for the ANN: distance to the nearest sensor and EMF exposure information recorded by nearest sensor. We assume 50 sensors are installed on selected street lamps denoted in green markers in Figure 10. 50% of the drive testing are selected as training data of the ANN, the remaining 50% are used for testing, which is used to produce results in Table 3.

In Table 3, MSE and R${}^{2}$ are used as the main metrics to evaluate the performance of ANN from different scenarios. Both MSE and R${}^{2}$ are obtained by averaging 10 times. Results produced from conventional and hybrid ANN are compared, where a significant improvement in terms of R${}^{2}$ can be found between conventional and hybrid ANN. Furthermore, the number of trainable parameters in Table 3 represents the parameters that could learn, which is proportional to execution time.

Figure 11 shows the performance of upper three scenarios in case 2 mentioned in Table 3, where we compare conventional ANN with proposed hybrid ANN. The closer blue and red dots get to the black diagonal, the more accurate predictions are. The training data and testing data have a similar degree of performance, which means there are no over-fitting problems. It is clear that dots are more concentrated in the figure on the right, which is from hybrid ANN. Figure 12 showing a Quantile-to-Quantile (Q-Q) plot from conventional ANN with proposed hybrid ANN as well. If targets and predictions are closer to each other, a linear relationship can be observed from the plotted quantiles. From all three figures, an overall linear relationship is observed with details improved around minimum values region in the figures from left to right. It aligns with our results in Table 3 and Figure 11.

Figure 13 shows a noiseless case, EMF exposure can be well reconstructed by using hybrid ANN, where a large majority of dots in blue are observed in the figures, showing relatively small error from predictions. Figure 14 tests the robustness of ANN in dealing with noisy data. The results are obtained using hybrid ANN as well, which corresponds to the last scenario in Table 3. After adding noise to data, the hybrid ANN is still able to predict EMF exposure with R${}^{2}$ = 0.78.

## 7. Discussion

There are two cases considered in Section 6, where case 1 only uses information from sensor networks and case 2 combines drive testing with aid from sensor networks. It is shown that a sufficient number of sensors are required to reconstruct spatial map of EMF exposure with good prediction. However, due to the cost of deploying real sensors, it is not ideal to adopt this method. While in the second case, only 50 sensors are required as extra information to assist the assessment of EMF exposure. The results show maximum R${}^{2}$ = 0.87 and R${}^{2}$ = 0.78 can be achieved with one-time drive testing in the noiseless and noisy scenarios. It shows the robustness of the ANN, that both cases with and without presence of noise can be assessed by ANN with good predictions.

Second, we prove that with new proposed hybrid ANN, better prediction of EMF exposure can be obtained for both cases. In case 2, we are able to assess the EMF exposure from certain frequency with R${}^{2}$ up to 0.87 under noiseless scenarios. It represents more than 87% of the targets can be explained using predictions from ANN. Furthermore, the hybrid ANN has better performance than the conventional ANN while less trainable parameters are required.

In the future work, real drive testing and sensor measurements will be used in the ANN to train and predict EMF exposure. Furthermore, an empirical based block-based PL model will be introduced, based on the real city structure, e.g., building height, width, road width. Since the current ANN work is based on the Tensorflow and Python environment, where GPU can boost the efficiency of computing, we will use GPU implemented on Nvidia Quadro RTX 8000 seed-Architecture Turin [29] in the future work. We believe that the methodology proposed in this paper is also applicable to 5G networks. The 5G network system model can be very different from normal 4G network, with new features, like massive multiple-input multiple-output (MIMO) and different frequency spectrum. However, the aggregating nature of EMF exposure from all effective BSA nearby will not change.

## 8. Conclusions

The time and spatial mapping of EMF exposure induced by BSA using ANN is studied in this paper. Sensor networks installed on street lamps in the 14th district of Pairs, drive testing and information of BSA which is publicly available are used to aid the reconstruction. The reference EMF exposure is built through simulations by considering a block-based stochastic PL model and time variation. We proposed a new efficient hybrid ANN structure and results show it outperforms the conventional ANN. First, reconstruction through sensor networks only containing different number of sensors are explored. We found that a quite large number of sensors is required for good reconstruction of EMF exposure map, which brings problems of high-cost in deployment and maintenance. Second, the reconstruction method through a combination of EMF sensors measurements and one-time drive testing has been employed. The results show that using a combination method, good prediction can be achieved with much fewer sensors required. Additional noise is considered to test the robustness of ANN. 

## Figures and Tables

**Figure 1 ijerph-17-03052-f001:**
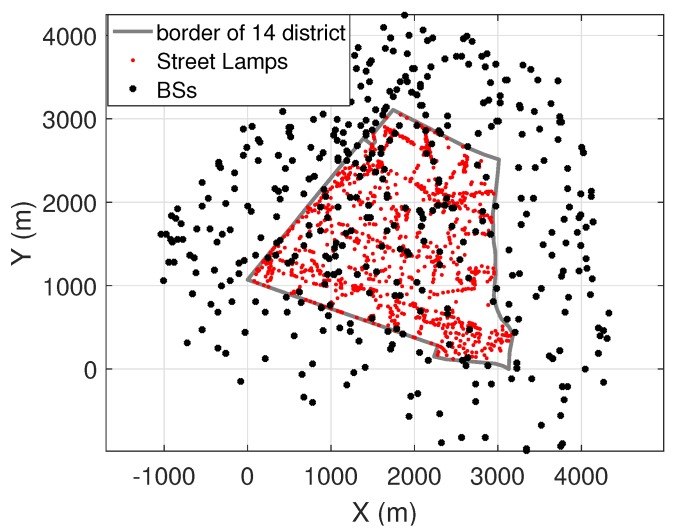
Base station antennas (BSA) and street lamps inside and around the 14th district in Paris.

**Figure 2 ijerph-17-03052-f002:**
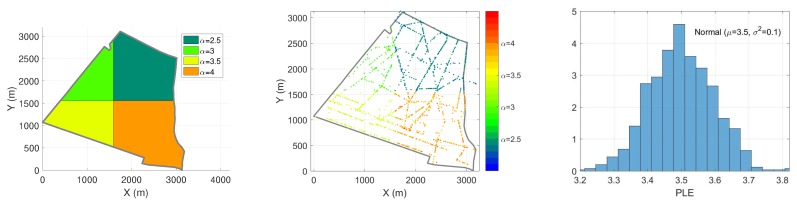
Block-based path loss (PL) model. (**Left**): Example of deterministic four-block based PLE model in the 14th district. (**Middle**): Map of the Block-based stochastic Path Loss Exponent (PLE) model. (**Right**): Receivers lie inside each block have normally-distributed PLE.

**Figure 3 ijerph-17-03052-f003:**
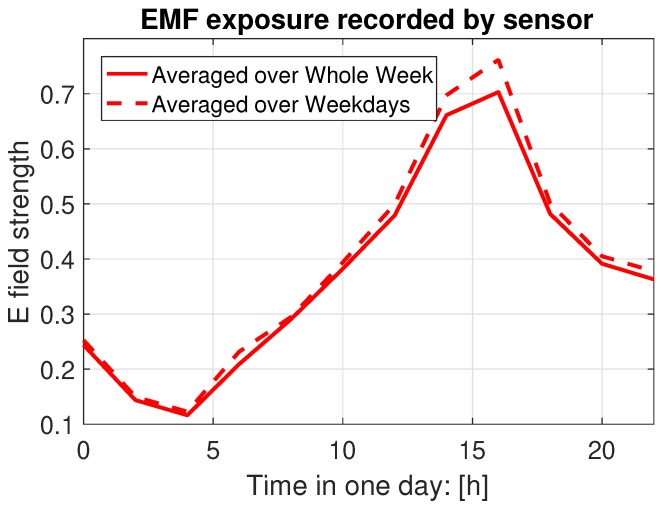
Time variation recorded by sensor network.

**Figure 4 ijerph-17-03052-f004:**
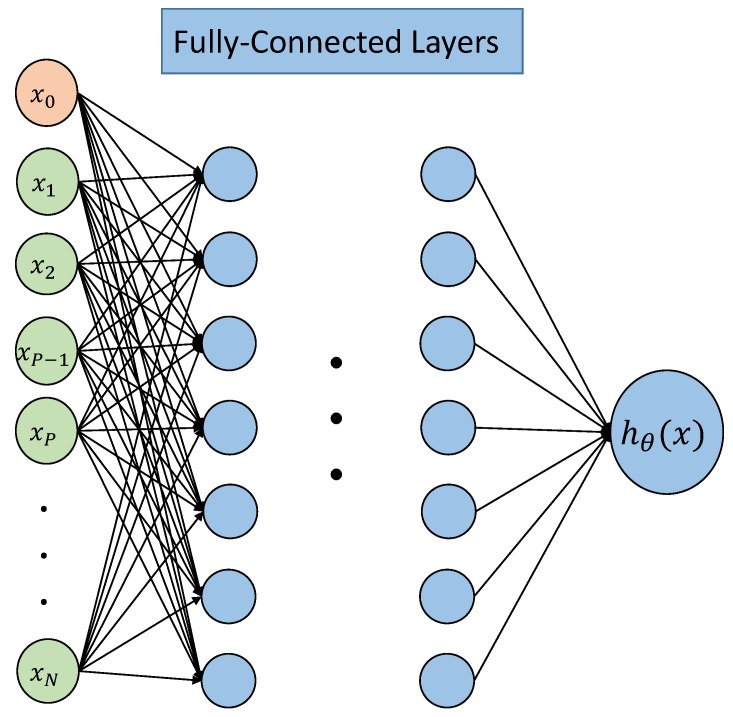
Structure of regression artificial neural networks (ANN).

**Figure 5 ijerph-17-03052-f005:**
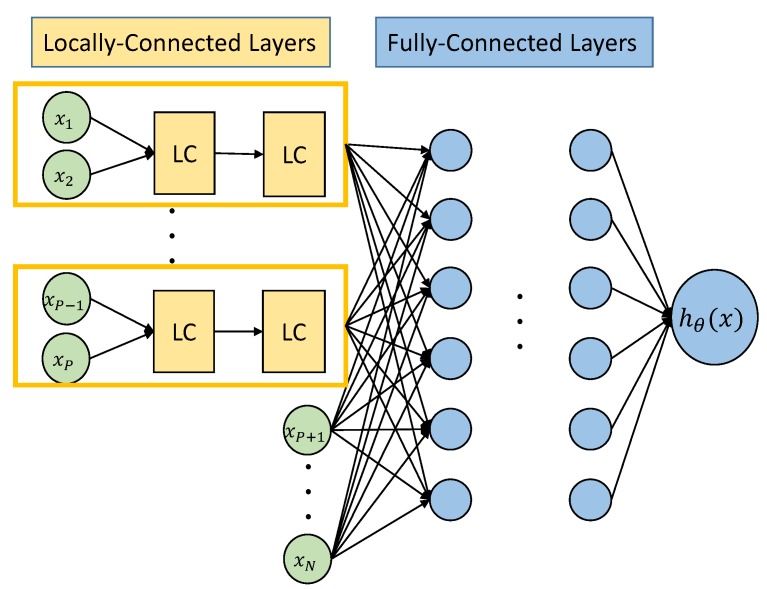
Structure of hybrid ANN.

**Figure 6 ijerph-17-03052-f006:**
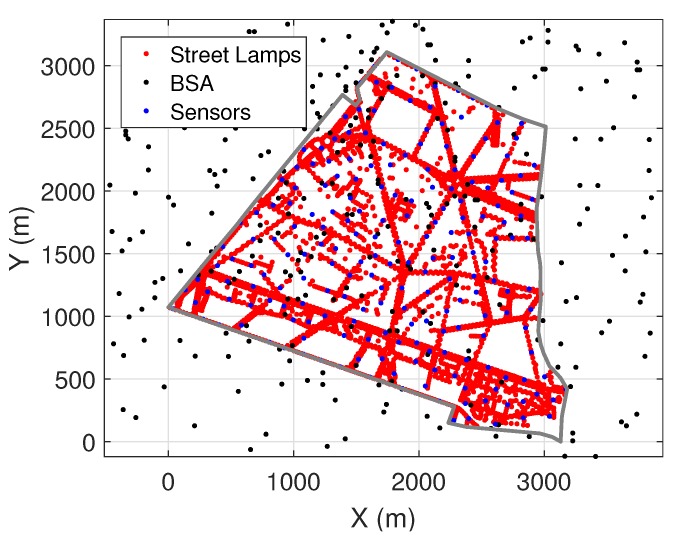
Sensor networks in the 14th district of Paris.

**Figure 7 ijerph-17-03052-f007:**
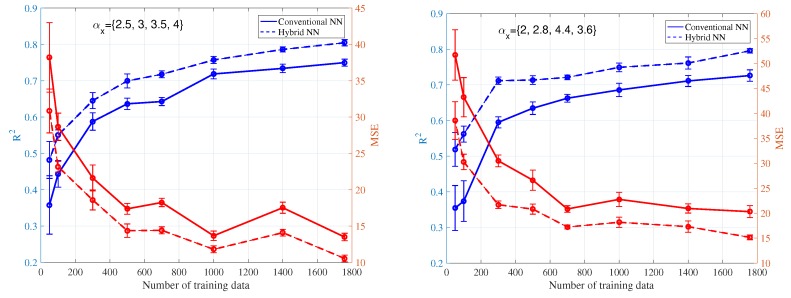
Averaged mean square error (MSE) and R${}^{2}$ with standard deviation (STD) based on sensor network inputs.

**Figure 8 ijerph-17-03052-f008:**
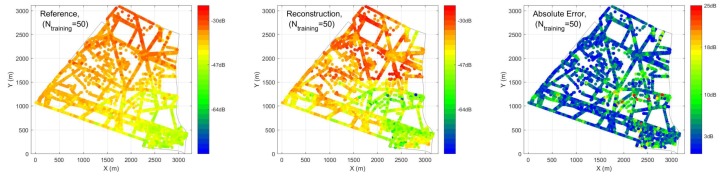
Exposure Reference Map (ERM), reconstructed electromagnetic field exposure map (EEM) and absolute error maps obtained from hybrid ANN for $N_{training}=50$ cases.

**Figure 9 ijerph-17-03052-f009:**
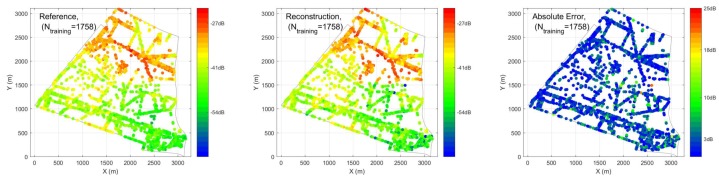
ERM, reconstructed EEM and absolute error maps obtained from hybrid ANN for $N_{training}=1758$ cases.

**Figure 10 ijerph-17-03052-f010:**
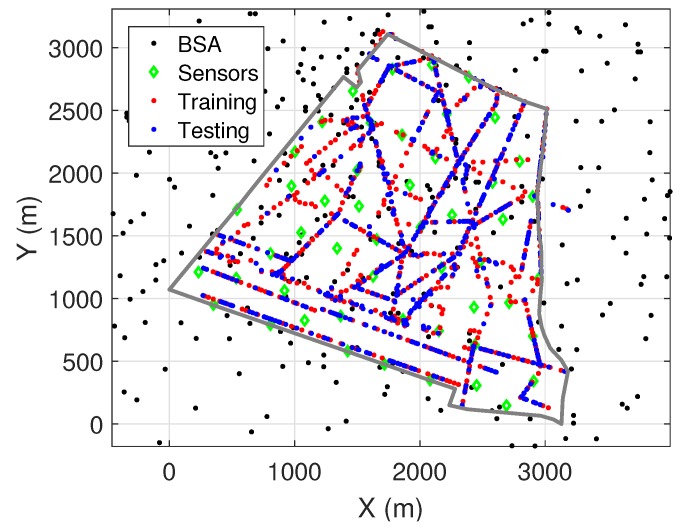
Selection of training and testing data.

**Figure 11 ijerph-17-03052-f011:**
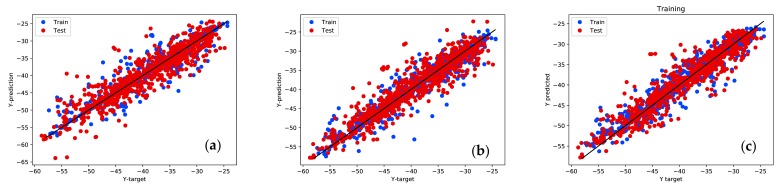
Scattering plot between predictions and targets from ANN. From left to right: (**a**) conventional ANN without considering sensor information as inputs of ANN; (**b**) conventional ANN considering sensor information as inputs of ANN; (**c**) hybrid ANN considering sensor information as inputs.

**Figure 12 ijerph-17-03052-f012:**
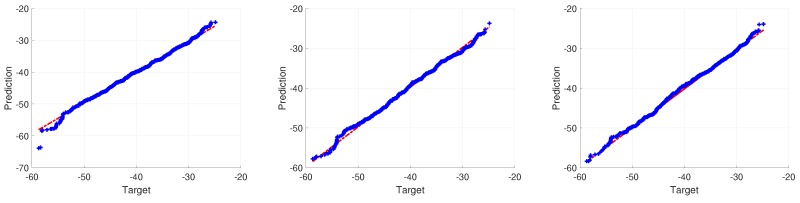
Quantile-to-Quantile (Q-Q) plot between predictions and targets from ANN. From left to right: (1) conventional ANN without considering sensor information as inputs of ANN; (2) conventional ANN considering sensor information as inputs of ANN; (3) hybrid ANN considering sensor information as inputs.

**Figure 13 ijerph-17-03052-f013:**
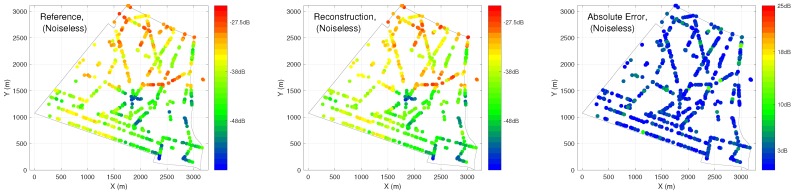
Spatial maps of EMF exposure obtained from hybrid ANN for noiseless case.

**Figure 14 ijerph-17-03052-f014:**
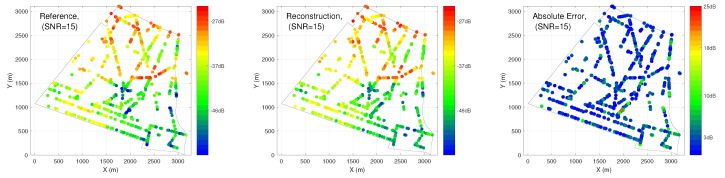
Spatial maps of EMF exposure obtained from hybrid ANN for noisy case (SNR = 15).

**Table 1 ijerph-17-03052-t001:** System setup.

Total number of sensors in case 1	3516
Number of sensors selected in case 2	50
Additive Noise level	SNR = 15dB
PLE in Block-based PL Model	$\alpha_{x}=\left\{2.5,3,3.5,4\right\}$ and {2,2.8,4.4,3.6}
$\sigma^{2}$ in normal distribution	$\sigma^{2}=\left\{0.1,0.15\right\}$
$N_{training}$, $N_{validation}$ and $N_{testing}$ in case 2	469, 201 and 670

**Table 2 ijerph-17-03052-t002:** Hyper-parameters in ANN.

Hyper-Parameters	Conv.-Case 1	Hybrid-Case 1	Conv.-Case 2	Hybrid-Case 2
Pre-processing	Standardization			
Num. of hidden layers	4	2(LC), 1(FC)	4	2(LC), 3(FC)
Num. of neurons	50	20(LC), 30(FC)	50	10(LC), 40(FC)
Learning rate	$1\times 10^{-4}$			
Batch size	10			
Patience in early stopping	30			

**Table 3 ijerph-17-03052-t003:** Performance of different ANN in case 2

Scenarios	Num. of Para.	MSE (±STD)	R${}^{2}$ (±STD)
Conventional ANN without considering sensors	8851	11.2 (±0.91)	0.81 (±0.02)
Conventional ANN considering sensors	8951	9.2 (±0.73)	0.85 (±0.01)
Hybrid ANN with considering sensors	5431	7.9 (±1.06)	0.87 (±0.02)
Conventional ANN without considering sensors with noise	8851	16.3(±0.85)	0.72 (±0.01)
Conventional ANN considering sensors with noise	8951	14.6 (±0.76)	0.75 (±0.01)
Hybrid ANN without considering sensors with noise	5431	12.8 (±0.96)	0.78 (±0.02)

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
