# Peer review of "Sensor-Aided EMF Exposure Assessments in an Urban Environment Using Artificial Neural Networks"

_ijerph, 2020, doi:10.3390/ijerph17093052_

Round 1
Reviewer 1 Report
In this paper, the author proposes a new hybrid-connected neural networks structure to study the reconstruction of EMF exposure, and compares it with conventional NN. The results show that the former has better performance and verified by simulation. But there are several points need to be addressed to improve the paper quality:
- Considering that the installed sensor networks may be affected by other factors such as bad weather, high-voltage transmission lines and substation equipment, how to analyze the impact of complex external interference on the system?
- The system model in this paper is based on the 4G cellular network. With the development of 5G technology, it will be popularized and applied soon. If the cellular network in the model is upgraded from 4G to 5G, whether the method proposed in this paper is still applicable, please explain.
- In the Stochastic Block-based Path Loss Model, the author says that the value of PLE around open space is close to 2. Similarly, Figure 2 shows the value of PLE in different areas. How to determine the value of PLE? Is there a calculation formula? Or is it based on empirical values?
- In Fig. 3, the author uses a trigonometric function to simulate time variation and obtain ft (t) in equation (2), but the accuracy and comparison of the simulation have not been given.
- What is R2 in Chapter 3? How is it defined? What is its expression? Please explain.
- There are no explanations for ANN training parameters, the number of hidden layer neurons and so on. And we know that the initial weights and thresholds in neural network have a certain impact on the network performance, how is it eliminated in the paper?
- In the end, the simulation analysis is given. How to implement the proposed hybrid ANN in practical applications? If it can be achieved, would it be a real-time model? How to get the data input to the neural network in real time? Need to set up wireless communication?
Author Response
We would like to thank the reviewer 1 for his or her thoughtful comments and efforts towards improving this manuscript. In the attched document, we address comments specific to reviewer 1.

Reviewer 2 Report
Dear Authors,
first of all thank you for your submitted manuscript.
The study presented by this manuscript is very interesting and has potential for future development/applications, if properly exploited. Hence, using GPU towards data processing capability is a wise proposal. Even though the 14th district of Paris isn't very variable topography-wise (from what I was able to understand), it might be an option to explore that data parameter as well, in the future.
Nevertheless, there are some minor corrections to be performed for it to be in proper publication shape. Comments are embedded on the attached document.
Regards.

Author Response
We would like to thank the reviewer 2 for his or her thoughtful comments and efforts towards improving this manuscript. The corrections have been taken into consideration in the revised manuscript. In the attched document, we address comments specific to reviewer 2.

Reviewer 3 Report
The english is really bad, and it needs extensive editing. It was difficult for me to follow through the paper.
As an example, just in the abstract:
- First sentence is too long.
- Sentence starting in line 9 makes no sense.
- Line 12 should probably start with an "and".
- Line 13, there's a comma and then the next word starts with a capital letter.
- Line 16 seems to be written using Google translate.
I don't have the time to go through the whole paper, but it needs extensive editing before being ready for publication.
Nevertheless, I like the topic in the way they describe it, and I think it is interesting to see how the antennas emit their signals through the landscape of a city.
Figure 4 is very basic. I would like to see their actual NN architecture, not a picture from Computer Science 101.
I think their idea of using locally connected neurons is interested. This is somewhat similar to CNN, which they mention briefly. I would like to see more about why their system is better over CNN.
I think their Figures plotting the data over the map of Paris are really good. And the results are sound.
Author Response
We would like to thank the reviewer 3 for his or her thoughtful comments and efforts towards improving this manuscript. In the attched document, we address comments specific to reviewer 3.

Round 2
Reviewer 1 Report
The present manuscript was revised and improved according to the given reviews and therefore improved highly in quality and acceptable.